# LEARNING TOP-k CLASSIFICATION WITH LABEL RANKING

## ABSTRACT

Class confusability and multi-label nature of examples inevitably arise in classification tasks with the increasing number of classes, which poses a huge challenge to classification. To mitigate this problem, top-$k$ classification is proposed, where the classifier is allowed to predict $k$ label candidates and the prediction result is considered correct as long as the ground truth label is included in the $k$ labels. However, existing top-k classification methods neglect the ranking of the ground truth label among the predicted $k$ labels, which has high application value. In this paper, we propose a novel three-stage approach to learn top-$k$ classification with label ranking. We first propose an ensemble based relabeling method and relabel the training data with $k$ labels, which is used to train the top-$k$ classifier. We then propose a novel top-$k$ classification loss function that aims to improve the ranking of the ground truth label. Finally, we have conducted extensive experiments on four text datasets and four image datasets, and the experimental results show that our method could significantly improve the performance of existing methods.

## 1 INTRODUCTION

Multi-class classification aims to classify examples into one of more than two classes, and as the number of classes increases to a large extent, e.g., thousands of classes, training a multi-class classifier will become extremely challenging due to the presence of multi-label nature of the examples and class confusability (Gupta et al., 2014; Lapin et al., 2015; Chang et al., 2017). To mitigate this problem, the task of top-$k$ classification is proposed (Berrada et al., 2018; Petersen et al., 2022), where the classifier is allowed to predict $k$ label candidates and the prediction result is considered correct as long as the ground truth label is included in the $k$ labels. This evaluation measure is commonly referred to as the top-$k$ error (Lapin et al., 2016), i.e., the loss function will not penalize $k-1$ mistakes. Though state-of-the-art models directly trained with cross-entropy can also yield remarkable results in terms of top-$k$ error, the data used for training must be both large and clean (Berrada et al., 2018), which cannot be guaranteed in real scenarios. Moreover, traditional top-1 error loss function like cross-entropy may have over-fitting problem when noisy label exists (Berrada et al., 2018). Hence, loss functions tailored for top-$k$ error minimization are needed.

However, existing top-$k$ classification loss functions (Lapin et al., 2015; Chang et al., 2017; Berrada et al., 2018) only consider whether the ground truth label is reported in the predicted $k$ labels and neglect the ranking of the ground truth label within the top-$k$ candidates. In fact, ranking is crucial for tasks of top-$k$ classification. For example, in a classic human-in-the-loop (Zanzotto, 2019) data annotation scenario, a classification model trained with a small amount of labeled data is used to predict for each unlabeled example, and humans are required to check the prediction result of a classification model and relabel those with low confidence. However, manually selecting the correct label from a large label set is time-consuming and inefficient. In this case, the model is allowed to predict the $k$ most likely labels so that humans could easily find the ground truth label from those $k$ labels, i.e., top-$k$ classification. Meanwhile, improving the ranking of the ground truth label in the $k$ labels will allow humans to get the ground truth label at the first time, which could effectively improve the efficiency of humans checking. The ranking motivation behind above scenario actually aligns with applications like recommendation system and search engine (Oosterhuis & de Rijke, 2020).

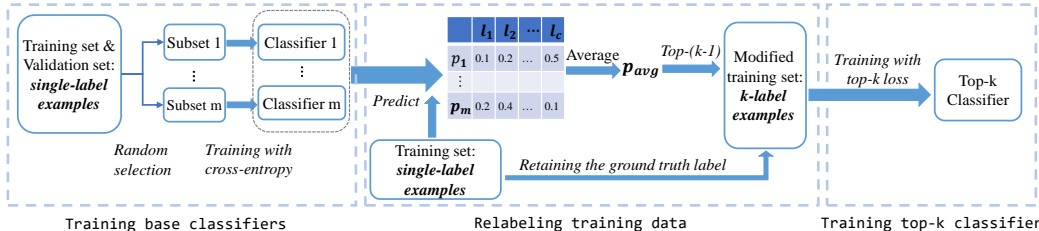

Figure 1: The proposed three-stage approach for top-$k$ classification.

Therefore, in this paper, we aim to design a top-$k$ classification method that can predict $k$ most likely labels where the ground truth label is not only included but also its ranking should be as high as possible. Driven by the idea of multi-label classification (MLC) (Tsoumakas & Katakis, 2007), we propose a novel three-stage approach for top-$k$ classification. As shown in Figure 1, in the first stage, we use the existing training and validation sets to train the base classifier which will be used to predict $k$-labels for each training example in next stage. Considering that the correct $k$ labels are crucial for top-$k$ classification in our problem setting, we use the idea of ensemble learning (Sagi & Rokach, 2018) to train $m$ base classifiers. Each base classifier is trained with classic cross-entropy loss function on the subset that randomly selected from training and validation sets. In the second stage, we relabel each single-label example to $k$-label example by using one ground truth label and $k - 1$ most likely labels. More specifically, we predict $m$ probability distributions $p$ for each sample by the $m$ base classifiers, and then average all $m$ probability distributions to get the average probability distribution $p_{avg}$. Finally, we output the most likely $k-1$ labels for each training sample besides its ground truth label according to the $p_{avg}$. We refer to these $k - 1$ most likely labels as pseudo-labels.

In the third stage, based on the transformed $k$-label examples, we train a multi-label classifier which predict exactly $k$ labels for a test example. Then, the trained multi-label classifier can be viewed as a top-$k$ classifier. It is important to note that, we propose a new top-$k$ loss function with label ranking (TkLR) for training in this stage. For the example with $k$ labels, TkLR aims to maximize the difference between the scores of these $k$ labels and the scores of other labels. To improve the ranking of the ground truth label, we embed an additional *rank loss* in TkLR, which aims to maximize the difference between the scores of the ground truth label and the scores of pseudo-labels.

Finally, we conduct sufficient experiments on four text datasets and four image datasets with BERT (Kenton & Toutanova, 2019) and Swin Transformer (Liu et al., 2021) as the backbone model respectively. Then we evaluate the experimental results with top-$k$ accuracy and Normalized Discounted Cumulated Gains at top $K$ ($NDCG@K$) (Wang et al., 2013). The experimental results demonstrate that our method could significantly improve the performance of existing top-$k$ classification methods.

In brief, the main contributions of this paper are summarized as follows:

- We propose to consider the ranking of ground truth label in top-$k$ classification, which has high application value but hasn't been well addressed so far.
- We propose a novel three-stage approach to learn top-$k$ classification with label ranking, which can be easily deployed with different classification models.
- We propose an ensemble based relabeling method to obtain the most likely $k$ labels for each training example which will benefit the final top-$k$ classification.
- We propose a novel top-$k$ loss function that takes the ranking of the ground truth label into account.
- The extensive experiments over different text and image datasets show that our method greatly outperforms existing baselines in terms of top-$k$ accuracy and $NDCG@K$ metrics.

The remainder of this paper is structured as follows. In Section 2, we present the related work. Section 3 describes our approach. We describe our experiments in Section 4. Section 5 concludes the paper.

## 2 RELATED WORK

In this section, we will present our related work in terms of top-$k$ classification and multi-label classification.

**Top-$k$ classification.** Lanpin et al. (Lapin et al., 2015) proposed a top-$k$ multiclass SVM based on a tight convex upper bound of the top-k hinge loss. Then Chu et al.(Chu et al., 2018) propose an optimized top-$k$ multiclass SVM algorithm, which employs semismooth Newton algorithm for the key building block to improve the training speed. Chang et al. (Chang et al., 2017) propose a generic, robust multiclass SVM formulation that directly aims at minimizing a weighted and truncated combination of the ordered prediction scores. Recently Berrada et al. (Berrada et al., 2018) propose Smooth loss function which is a new top-$k$ classification loss function for deep neural networks. The Smooth loss function creates a margin between the correct top-$k$ predictions and the incorrect ones. Petersen et al. (Petersen et al., 2022) propose a family of differentiable top-$k$ cross-entropy classification losses and relax the assumption of a fixed k. However, all the above methods do not take into account the ranking of ground truth labels among the predicted $k$ labels.

**Multi-label classification.** Boutell et al. (Boutell et al., 2004) decomposed the multi-label classification into several independent binary classification problems. Read et al.(Read et al., 2011) transformed the multi-label classification problem into a chain of binary classification problems. For text dataset, Yang et al. (Yang et al., 2018) proposed a sequence-to-sequence multi-label classification model to learn the correlation among labels. Huang et al. (Huang et al., 2021) introduce the application of balancing loss functions for multi-label text classification. For image dataset, Chen et al. (Chen et al., 2019) propose a multi-label classification model based on Graph Convolutional Network (GCN) to capture and explore label dependencies. Lanchantin et al. (Lanchantin et al., 2021)propose the Classification Transformer (C-Tran), a general framework for multi-label image classification that leverages Transformers to exploit the complex dependencies among visual features and labels.

Though our method is inspired from multi-label classification, the difference is obvious: 1) The input is different. The number of ground truth label per training sample for multi-label classification is indeterminate, while the training sample of our method only has one ground truth label. 2) The output result is different. The number of labels predicted by multi-label classification is also uncertain, while the number of labels predicted by our method is fixed at $k$. 3) The evaluation measure is different. The all predicted labels of multi-label classification are required to be ground truth labels, while our method only requires the ground truth label included in the predicted $k$ labels.

## 3 THREE-STAGE APPROACH

### 3.1 OVERVIEW

As can be seen from the Figure1, three stages of our approach are: training base classifiers, relabeling training data and training top-$k$ classifier.

**Stage 1: Training base classifiers.** Base classifiers are used to predict the $k$ most likely labels for each training example in stage 2. Considering that the correct $k$ labels are crucial for top-$k$ classification in our problem setting, we use the idea of ensemble learning to train $m$ base classifiers. Specifically, given the single-label dataset $D$ which is composed of training set and validation set. we randomly select $m$ different subsets from dataset $D$ to train the $m$ base classifiers, where the proportion of subsets in dataset $D$ is $\alpha$. The loss function of training process is cross-entropy loss function. The settings of hyperparameters $\alpha$ and $m$ will be discussed in Appendix 4.2.

**Stage 2: Relabeling training data.** The purpose of this step is to relabel each single-label training example with its $k$ most likely labels. We consider the ground truth label to be the most likely label, so we only need to find the other $k-1$ most likely labels. Specifically, given the single-label training set $\{(x_i, y_i)\}_{i=1}^t$, where $x_i$ are the input features of each example $i$ and $y_i$ is the ground truth label. We aim to relabel each example $(x_i, y_i)$ to $(x_i, Y_i)$, where $Y_i \in \mathbb{R}^k$ is composed of the ground truth label $y_i$ and the $k-1$ other labels that are most related to $x_i$. To this end, we use the $m$ base classifiers to predict $m$ different probability distribution $p$ of all labels for each sample in the training set, where $p \in \mathbb{R}^c$, $c$ is the number of total labels. Then we average the $m$ probability

distributions and get the average probability distribution $p_{avg}$. According to the $p_{avg}$, we output the $k-1$ labels with the highest probability except the ground truth label, and we use these $k-1$ labels as the pseudo-labels of the sample. At this point, relabeling is complete, and each example in the training set has one ground truth label and $k-1$ pseudo-labels. Note that, the training set in this stage is used for deriving a probability distribution over different labels and obtaining for a modified training set.

**Stage 3: Training top-$k$ classifier.** We take the output of the second stage as training data and use our proposed top-$k$ loss function to train a new top-$k$ classification model. The loss function is described in Section 3.2.

## 3.2 TOP-K LOSS FUNCTION WITH LABEL RANKING

In this section, we will introduce our rank top-$k$ loss function. In order to better understand the top-$k$ classification, we start with the simple case of $k$=1, then the top-1 classification is an ordinary multi-class classification. Given a training sample $(x_i, y_i)$, $i = 1, \ldots, t$, where $y \in \mathcal{Y} := \{1, \ldots, c\}$, the classifier will output the score of the sample $x$ on each label, i.e., the score vector $\mathbf{s} := \{s_1, \ldots, s_c\}$. The multi-class classification aims to maximize the probability $p_y$ of ground truth label $y$, where $p_y$ is calculated by Softmax. Then multi-class classification aims at minimizing the following Softmax cross-entropy loss:

$$\mathcal{L}_{k=1} = -\log(\frac{e^{s_y}}{\sum_{j=1}^{c} e^{s_j}}) \tag{1}$$

where $s_y$ is the score of the ground truth label. From Eq.1, it can be seen that reducing the loss is equivalent to increasing $s_y$. Taking $s_y$ as the object, Eq.1 can be transferred as follows:

$$\begin{aligned} \mathcal{L}_{k=1} &= -\log(\frac{e^{s_y}}{\sum_{j=1}^{c} e^{s_j}}) = -\log \frac{1}{\sum_{j=1}^{c} e^{s_j - s_y}} \\ &= \log \sum_{j=1}^{c} e^{s_j - s_y} = \log \left( 1 + \sum_{j=1, j \neq y}^{c} e^{s_j - s_y} \right) \end{aligned} \tag{2}$$

The goal of Eq.2 is to minimize $(s_j - s_y)$, that is, it is hoped that the score of the ground truth label should be much larger than the scores of other labels. We define the ground truth label as positive label and the rest as negative labels, then Eq.2 can be rewritten as follows:

$$\mathcal{L}_{k=1} = \log \left( 1 + \sum_{n \in \boldsymbol{neg}} e^{s_n - s_p} \right) \tag{3}$$

where $\boldsymbol{neg}$ is the set of the negative labels, $s_n$ is the score of the negative label, $s_p$ is the score of the positive label. Since $k = 1$, Eq.3 has 1 positive label and $c - 1$ negative labels. Obviously, Eq.3 takes maximizing the score of positive label as the optimization goal, and directly outputs the label with the largest score as the prediction result in the prediction stage.

When $k > 1$, each training sample has $k$ labels, like the case of $k = 1$, the loss function expects that the scores of the $k$ labels are larger than the rest labels. Specifically, given a training sample $(x_i, Y_i)$ which is the top-$k$ relabeling result of $(x_i, y_i)$, where $Y_i \in \mathbb{R}^k$ is composed of the ground truth label $y_i$ and the $k-1$ other pseudo-labels that are most related to $x_i$, the loss function aims to maximize the scores of the $Y_i$ and minimize the scores of the rest labels. Similarly, we define $Y_i$ as positive labels and the rest as negative labels, then loss function can be rewritten as follows:

$$\mathcal{L}_{k>1} = \log(1 + \sum_{n \in \boldsymbol{neg}, p \in \boldsymbol{pos}} e^{s_n - s_p}) \tag{4}$$

where $\boldsymbol{pos}$ is the set of the $k$ positive labels and $\boldsymbol{neg}$ is the set of the $c-k$ negative labels. Obviously, Eq.4 hopes to maximize the score of positive labels, and the classifier only needs to output the $k$ labels with the largest scores as the predicted results.

Table 1: Statistics of the datasets

| Dataset | Train | Validate | Test | Labels | Domain |
|---|---|---|---|---|---|
| Ohsumed | 3357 | 2026 | 2017 | 23 | Text |
| WOS | 28139 | 9451 | 9395 | 134 | Text |
| 20Ng | 11270 | 3692 | 3820 | 20 | Text |
| TREC | 4882 | 570 | 500 | 50 | Text |
| CIFAR100 | 40000 | 10000 | 10000 | 100 | Image |
| Aircraft | 3334 | 3333 | 3333 | 100 | Image |
| CUB-200-2011 | 7099 | 2315 | 2374 | 200 | Image |
| Indoor67 | 9398 | 3069 | 3153 | 67 | Image |

However, the loss function in Eq.4 does not consider the ranking of the ground truth label. Another goal of our top-$k$ classification is to expect that the ground truth label has a higher ranking in the predicted $k$ labels, which means that the score of the ground truth label should be as large as possible. More specifically, the score of the ground truth label $s_y$ should be larger than that of other positive labels, i.e., maximize $(s_y - s_p), p \neq y$. Then the final loss function can be written as follows:

$$
\begin{aligned}
\mathcal{L} &= \log(1 + \sum_{n \in \textbf{\textit{neg}}, p \in \textbf{\textit{pos}}} e^{s_n - s_p} + \sum_{p \in \textbf{\textit{pos}} \setminus y} e^{s_p - s_y}) \\
&= \log(1 + \sum_{n \in \textbf{\textit{neg}}} e^{s_n} \sum_{p \in \textbf{\textit{pos}}} e^{-s_p} + e^{-s_y} \sum_{p \in \textbf{\textit{pos}} \setminus y} e^{s_p})
\end{aligned}
\tag{5}
$$

where $\textbf{\textit{pos}} \setminus y$ is the set of positive labels other than the ground truth label $y$. We name the third part $e^{-s_y} \sum_{p \in \textbf{\textit{pos}} \setminus y} e^{s_p}$ in the loss function of Eq.5 as *rank loss*.

Adding *rank loss* to the final loss function has two benefits. The first is to improve the ranking of the ground truth label in the $k$ predicted labels mentioned above. Second, the top-$k$ classification expects the ground truth label to appear in the $k$ predicted labels, and the *rank loss* could improve the score of the ground truth label, which is helpful for the ground truth label to be predicted in the $k$ labels more likely. Therefore, *rank loss* also helps to improve the accuracy of top-$k$ classification.

## 4 EXPERIMENT

In this section, we perform extensive experiments to validate the effectiveness of our method. First, we describe the experimental setup in detail. Second, we compare our method with different approaches. We then conduct experiments on both text datasets and image datasets, simultaneously evaluating our experimental results with two evaluation metrics. The source code will be available at https://github.com/Tracy-6914/TkLR

### 4.1 EXPERIMENTAL SETUP

**Dataset and model selection.** In our experiment, we select 4 single-label text datasets and 4 single-label image datasets: Ohsumed (Joachims, 1998), 20NG (Johnson & Zhang, 2016), WOS (Kowsari et al., 2017), TREC (Li & Roth, 2002), CIFAR100 (Krizhevsky et al., 2009), Aircraft (Lu et al., 2021), CUB-200-2011 (Wah et al., 2011), Indoor67 (Quattoni & Torralba, 2009). Ohsumed includes the medical abstracts of MEDLINE database and describes different cardiovascular diseases. 20 Newsgroups (20NG) is a collection of newsgroup documents posted on 20 different topics. WOS-46985 (WOS) collects the abstracts of papers published in Web Of Science. The TREC dataset is dataset for question classification consisting of open-domain, fact-based questions divided into broad semantic categories. CIFAR100 is a subset of the Tiny Images dataset and consists of 60000 32x32 color images. CUB-200-2011 is a challenging dataset of 200 bird species. Aircraft contains different aircraft model variants, most of which are airplanes. Indoor67 contains 67 indoor scene categories, and a total of 15620 images. Table 1 shows the statistics of all datasets. Our method is a general top-$k$ classification method that can be applied to various models. In our experiment, we choose BERT (Kenton & Toutanova, 2019) as the backbone model for the text dataset and Swin Transformer (Liu et al., 2021) as the backbone model for image dataset.

**Comparison methods.** For text dataset, we compare our method with four approaches: (1) **BERT + CE**, a simple multi-class classification method that uses cross-entropy (CE) as the loss function of BERT. (2) **BERT+Top-k SVM**, which obtains the features from the fine-tuned BERT and then input the features into top-$k$ SVM (Lapin et al., 2015) for top-$k$ classification. (3) **BERT+Top-k Entropy**, which trains the top-$k$ classifier by top-$k$ Entropy (Lapin et al., 2015) and the features obtained from the fine-tuned BERT. (4) **BERT+Top-k SVM**$_{Semismooth}$, which use Top-k SVM$_{Semismooth}$ (Chu et al., 2018) and the features obtained from the fine-tuned BERT to train the top-$k$ classifier. (5) **BERT + Smooth**, which uses the recent top-$k$ loss function Smooth loss function (Berrada et al., 2018) as the loss function of the BERT. (6) **BERT + Relabel**, a multi-label classification method of BERT that uses the training set after relabeling process and uses the binary cross-entropy loss function. (7) **BalancedLoss + Relabel**, the state-of-the-art multi-label text classification model BalancedLoss (Huang et al., 2021) trained by the training set after relabeling process.

Similarly, we compare our method with the following approaches on image dataset: (1) **Swin + CE**, a multi-class classification method that uses cross-entropy (CE) as the loss function of Swin Transformer. (2) **Swin+Top-k SVM**, which obtains the features from the fine-tuned Swin and then input the features into top-$k$ SVM (Lapin et al., 2015) for top-$k$ classification. (3) **Swin+Top-k Entropy**, which trains the top-$k$ classifier by top-$k$ Entropy (Lapin et al., 2015) and the features obtained from the fine-tuned Swin. (4) **Swin+Top-k SVM**$_{Semismooth}$, which use Top-k SVM$_{Semismooth}$ (Chu et al., 2018) and the features obtained from the fine-tuned Swin to train the top-$k$ classifier. (5) **Swin + Smooth**, which uses the Smooth loss function as the loss function of the Swin Transformer (6) **Swin + Relabel**, a multi-label classification method for Swin Transformer that uses the training set after relabeling process and uses the binary cross-entropy loss function. (7) **C-Tran + Relabel**, the state-of-the-art multi-label image classification model C-Tran (Lanchantin et al., 2021) trained by the training set after relabeling process.

**Ablation experiment.** We simply name our method as **BERT (Swin) + Relabel + TkLR**, and to demonstrate the effectiveness of our proposed *rank loss*, we remove the *rank loss* from our method and conducted experiment, namely **BERT (Swin) + Relabel + TkLR-No-Rank**.

**Experimental setting.** In the base classifiers training process, we choose BERT as the backbone network of the classifier for text dataset and Swin-small as the backbone network of the classifier for image dataset. In the top-$k$ classifier training process, for text dataset, we set epoch to 20, maximum sequence length to 512, batch size to 10, and learning rate to 2e-5. We train our model with AdamW (Loshchilov & Hutter, 2017) optimizer with weight decay = 0.01. For image classification, we set epoch to 100, batch size to 64, learning rate to 5e-5 and adopt AdamW optimizer with weight decay = 5e-2. Training process and convergence analyses of the loss function are discussed in Appendix A.2

**Evaluation metrics.** Considering the ranking of the ground true label among the predicted $k$ labels, we use top-$k$ accuracy (Yan et al., 2018) and Normalized Discounted Cumulated Gains at top $K$ ($NDCG@k$) (Wang et al., 2013) as our evaluation metrics. The specific calculation methods are as follows.

Top-$k$ accuracy is defined as:

$$Acc = \sum_{i=1}^{n} \frac{flag_i}{n}, \qquad flag_i = \begin{cases} 1 & \text{If ground truth label in the predicted } k \text{ labels} \\ 0 & \text{Otherwise} \end{cases} \tag{6}$$

where $n$ is the number of samples.

$NDCG@k$ is defined as:

Suppose that $l$ is the total number of the labels, $NDCG@K$ is defined according to the predicted score vector $\hat{y} \in \mathbb{R}^l$ and the ground truth label vector $y \in \{0,1\}^l$ as follows:

$$DCG@k = \sum_{j \in \text{rank}_k(\hat{y})} \frac{y_j}{\log(pos(j)+1)}, \qquad NDCG@k = \frac{DCG@k}{\sum_{i=1}^{\min(k,\|y\|_0)} \frac{1}{\log(i+1)}} \tag{7}$$

where $\text{rank}_k(y)$ is the label indexes of the top-$k$ highest scores of the current prediction result, $pos(j)$ is the position of $j$, i.e., $pos(j) = 1, 2, 3...$ $\|y\|_0$ counts the number of labels in the ground truth label vector $y$.

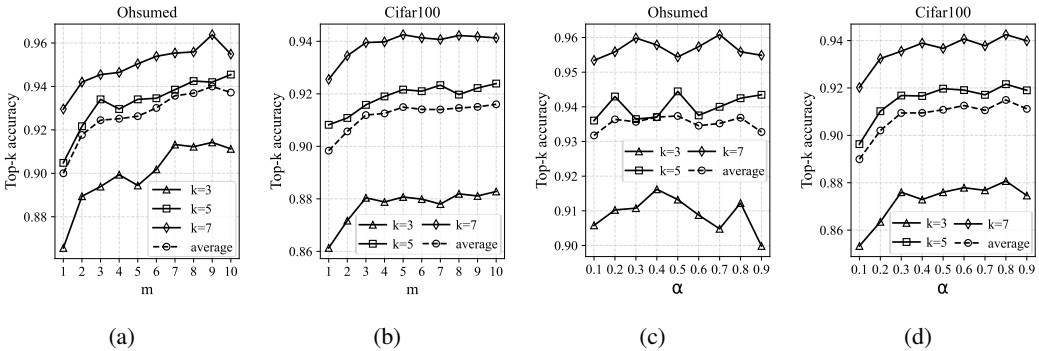

Figure 2: Setting of hyperparameters $\alpha$ and $m$

Table 2: Top-$k$ accuracy results on text datasets

| Methods | Ohsumed | | | 20NG | | | WOS | | | TREC | | |
|---|---|---|---|---|---|---|---|---|---|---|---|---|
| | Top 3 | Top 5 | Top 7 | Top 3 | Top 5 | Top 7 | Top 3 | Top 5 | Top 7 | Top 3 | Top 5 | Top 7 |
| BERT+CE | 0.8567 | 0.8984 | 0.9207 | 0.9623 | 0.9812 | 0.9812 | 0.8923 | 0.9187 | 0.9327 | 0.9680 | 0.9780 | 0.9800 |
| BERT+Top-k SVM | 0.8691 | 0.9048 | 0.9296 | 0.9702 | 0.9851 | 0.9898 | 0.8879 | 0.9139 | 0.9308 | 0.9500 | 0.9620 | 0.9780 |
| BERT+Top-k Entropy | 0.8820 | 0.9103 | 0.9291 | 0.9715 | 0.9851 | 0.9906 | 0.8956 | 0.9167 | 0.9280 | 0.9360 | 0.9440 | 0.9260 |
| BERT+Top-k $\text{SVM}_{Semismooth}$ | 0.8840 | 0.9182 | 0.9390 | 0.9699 | 0.9859 | 0.9908 | 0.9046 | 0.9275 | 0.9386 | 0.9500 | 0.9720 | 0.9780 |
| BERT+Smooth | 0.8587 | 0.8855 | 0.9103 | 0.9573 | 0.9712 | 0.9838 | 0.8978 | 0.9137 | 0.9218 | 0.9620 | 0.9660 | 0.9720 |
| BERT+Relabel | 0.8954 | 0.9316 | 0.9504 | 0.9673 | 0.9835 | 0.9898 | 0.9095 | 0.9348 | 0.9494 | 0.9760 | 0.9800 | 0.9860 |
| BalancedLoss+Relabel | 0.8780 | 0.9236 | 0.9484 | 0.9631 | 0.9822 | 0.9890 | 0.9088 | 0.9343 | 0.9485 | 0.9680 | 0.9820 | 0.9820 |
| BERT+Relabel +TkLR-No-Rank | 0.8984 | 0.9266 | 0.9514 | 0.9699 | 0.9817 | 0.9885 | 0.9118 | 0.9350 | 0.9485 | 0.9760 | 0.9820 | 0.9800 |
| **BERT+Relabel+TkLR** | **0.9147** | **0.9415** | **0.9579** | **0.9764** | **0.9882** | **0.9927** | **0.9191** | **0.9402** | **0.9564** | **0.9780** | **0.9820** | **0.9860** |

All experiments were implemented with Python 3.8 and PyTorch 1.8 on a Linux server. Our experiments for text classification on the RTX 3080Ti and for image classification on the RTX 3090.

## 4.2 HYPERPARAMETER SETTING

In our experiments, we employ Bagging (Zhou, 2012) method to train $m$ base classifiers. The Bagging method randomly selects data with a proportion of $\alpha$ from the total training data for training each time. We conduct experiments both on text and image datasets to determine the number of base classifiers $m$ and proportion $\alpha$. We first randomly initialize $\alpha$, and determine the optimal $m$ with the fixed $\alpha$. Then determine $\alpha$ with the optimal $m$. The experimental results are shown in Figure 2. Figure 2a and Figure 2b show that in general, the larger $m$ is, the higher the top-$k$ accuracy is. On the Ohsumed dataset, when $m = 8$, the overall results are good, and the performance is no longer significant when $m$ continues to increase. On the Cifar100 dataset, the result is also good when $m = 5$, and the effect of increasing $m$ is not obvious. Figure 2c and Figure 2d show the performance of different $\alpha$ with the optimal $m$. The results show that when $\alpha = 0.8$, top-$k$ classifier can achieve the best performance on the Ohsumed dataset and the cifar100 dataset. We found similar experimental results on other datasets. Therefore, for hyperparameter $m$, we set the $m$ to 8 for the text dataset and $m$ to 5 for the image dataset. For hyperparameter $\alpha$, we set $\alpha$ to 0.8 for both the text dataset and the image dataset. The theoretically analysis of hyperparameters is present in A.1

## 4.3 EXPERIMENTAL RESULT ON TEXT DATASET

Table 2 and Table 3 show the top-$k$ accuracy and the $NDCG$ values for different $k$ of all methods on text dataset, respectively. It is clear that our method BERT+Relabel+TkLR achieves the best results in both top-$k$ accuracy and $NDCG$ metrics, which demonstrates the effectiveness of our method to solve the top-$k$ classification problem.

As can be seen from the top-$k$ accuracy in Table 2, our method is much better than the method BERT+CE. Especially on Ohsumed dataset, when set different $k$, the accuracy of our method is at

Table 3: $NDCG@k$ results on text datasets

| Methods | Ohsumed | | | 20NG | | |
|---|---|---|---|---|---|---|
| | NDCG@3 | NDCG@5 | NDCG@7 | NDCG@3 | NDCG@5 | NDCG@7 |
| BERT+CE | 0.8016 | 0.8188 | 0.8265 | 0.9230 | 0.9307 | 0.9332 |
| BERT+Top-k SVM | 0.8101 | 0.8229 | 0.8269 | 0.9239 | 0.9118 | 0.9083 |
| BERT+Top-k Entropy | 0.8249 | 0.8339 | 0.8376 | 0.9260 | 0.9279 | 0.9280 |
| BERT+Top-k SVM$_{Semismooth}$ | 0.8274 | 0.8407 | 0.8458 | 0.9283 | 0.9336 | 0.9342 |
| BERT+Smooth | 0.8021 | 0.8064 | 0.8037 | 0.8948 | 0.8879 | 0.8897 |
| BERT+Relabel | 0.7731 | 0.7988 | 0.7536 | 0.8196 | 0.7891 | 0.6393 |
| BalancedLoss+Relabel | 0.7700 | 0.7794 | 0.7787 | 0.8742 | 0.8315 | 0.8456 |
| BERT+Relabel+TkLR-No-Rank | 0.7773 | 0.7739 | 0.7283 | 0.8658 | 0.7254 | 0.6661 |
| **BERT+Relabel+TkLR** | **0.8401** | **0.8508** | **0.8528** | **0.9342** | **0.9393** | **0.9372** |
| Methods | WOS | | | TREC | | |
| | NDCG@3 | NDCG@5 | NDCG@7 | NDCG@3 | NDCG@5 | NDCG@7 |
| BERT+CE | 0.8579 | 0.8688 | 0.8737 | 0.9465 | 0.9507 | 0.9514 |
| BERT+Top-k SVM | 0.8569 | 0.8665 | 0.8709 | 0.9347 | 0.9332 | 0.9334 |
| BERT+Top-k Entropy | 0.8645 | 0.8700 | 0.8725 | 0.9132 | 0.9201 | 0.8972 |
| BERT+Top-k SVM$_{Semismooth}$ | 0.8705 | 0.8798 | 0.8836 | 0.9331 | 0.9413 | 0.9439 |
| BERT+Smooth | 0.8550 | 0.8513 | 0.8487 | 0.9396 | 0.9326 | 0.9359 |
| BERT+Relabel | 0.7793 | 0.7833 | 0.8115 | 0.8984 | 0.8192 | 0.7370 |
| BalancedLoss+Relabel | 0.8285 | 0.8348 | 0.8323 | 0.8942 | 0.8962 | 0.6870 |
| BERT+Relabel+TkLR-No-Rank | 0.7953 | 0.7792 | 0.7935 | 0.9024 | 0.8376 | 0.7452 |
| **BERT+Relabel+TkLR** | **0.8768** | **0.8809** | **0.8840** | **0.9474** | **0.9517** | **0.9523** |

Table 4: Top-$k$ accuracy results on image datasets

| Methods | Cifar100 | | | Aircraft | | | CUB-200-2011 | | | Indoor67 | | |
|---|---|---|---|---|---|---|---|---|---|---|---|---|
| | Top 3 | Top 5 | Top 7 | Top 3 | Top 5 | Top 7 | Top 3 | Top 5 | Top 7 | Top 3 | Top 5 | Top 7 |
| Swin+CE | 0.8694 | 0.9117 | 0.9330 | 0.8797 | 0.9142 | 0.9316 | 0.9385 | 0.9613 | 0.9726 | 0.9483 | 0.9696 | 0.9822 |
| Swin+Top-k SVM | 0.8592 | 0.9050 | 0.9277 | 0.8974 | 0.9307 | 0.9499 | 0.9364 | 0.9650 | 0.9730 | 0.9353 | 0.9670 | 0.9772 |
| Swin+Top-k Entropy | 0.8570 | 0.9029 | 0.9259 | 0.8695 | 0.8986 | 0.9100 | 0.8614 | 0.9254 | 0.9465 | 0.9198 | 0.9312 | 0.9543 |
| Swin+Top-k SVM$_{Semismooth}$ | 0.8596 | 0.9067 | 0.9289 | 0.8968 | 0.9298 | 0.9499 | 0.9398 | 0.9667 | 0.9743 | 0.9423 | 0.9667 | 0.9791 |
| Swin+Smooth | 0.8741 | 0.9133 | 0.9326 | 0.8905 | 0.9316 | 0.9424 | 0.9495 | 0.9676 | 0.9760 | 0.9413 | 0.9619 | 0.9708 |
| Swin+Relabel | 0.8559 | 0.9077 | 0.9335 | 0.8644 | 0.9178 | 0.9391 | 0.9196 | 0.9549 | 0.9747 | 0.9340 | 0.9689 | 0.9784 |
| C-Tran+Relabel | 0.7810 | 0.8564 | 0.8877 | 0.8470 | 0.8962 | 0.9259 | 0.9073 | 0.9423 | 0.9604 | 0.8979 | 0.9407 | 0.9629 |
| Swin+Relabel +TkLR-No-Rank | 0.8612 | 0.9097 | 0.9345 | 0.8773 | 0.9181 | 0.9388 | 0.9410 | 0.9638 | 0.9752 | 0.9445 | 0.9705 | 0.9826 |
| **Swin+Relabel+TkLR** | **0.8830** | **0.9265** | **0.9442** | **0.9040** | **0.9358** | **0.9511** | **0.9528** | **0.9718** | **0.9832** | **0.9531** | **0.9759** | **0.9838** |

least 3.7% higher than that of BERT+CE method. At the same time, cross entropy aims to maximize the probability of the ground truth label, hence BERT+CE will perform well on the $NDCG$ metric in Table 3. Nevertheless, the results of our method on $NDCG$ metric are still better than that of BERT+CE, which illustrates the effectiveness of our *rank loss*. The top-$k$ accuracy results of BERT+Relabel are similar to the results of BalancedLoss+Relabel. Besides, in the most cases, the results of BERT+Relabel and BalancedLoss+Relabel are better than BERT+CE, BERT+Smooth, BERT+Top-k SVM, BERT+Top-k Entropy and BERT+Top-k SVM$_{Semismooth}$ overall on all text datasets, which illustrate the relabeling process could improve the top-$k$ classification accuracy and the idea of predicting the most likely $k$ labels is effective. However, the results of BERT+Relabel and BalancedLoss+Relabel on $NDCG$ metric are bad, because they do not consider the ranking of the ground truth label, on the contrary, our loss function does.

Compared with BERT+Relabel, our method BERT+Relabel+TkLR achieves better results in top-$k$ accuracy and $NDCG$ metric on all datasets, which shows that our proposed loss function is very effective. Though the top-7 accuracy of BERT+Relabel is equal to BERT+Relabel+TkLR, the result on $NDCG$ metric of BERT+Relabel+TkLR are extremely better than BERT+Relabel, which shows the strength of *rank loss*. Comparing the results of BERT+Relabel+TkLR-No-Rank and BERT+Relabel+TkLR shows that adding *rank loss* can not only improve the top-$k$ accuracy, but also improve the $NDCG$ metric.

Table 5: $NDCG@k$ results on image datasets

| Methods | Cifar100 | | | Aircraft | | |
|---|---|---|---|---|---|---|
| | NDCG@3 | NDCG@5 | NDCG@7 | NDCG@3 | NDCG@5 | NDCG@7 |
| Swin+CE | 0.8029 | 0.8204 | 0.8278 | 0.8111 | 0.8253 | 0.8313 |
| Swin+Top-k SVM | 0.7960 | 0.8149 | 0.8232 | 0.8332 | 0.8458 | 0.8507 |
| Swin+Top-k Entropy | 0.7932 | 0.8136 | 0.8228 | 0.7541 | 0.8125 | 0.8121 |
| Swin+Top-k SVM$_{Semismooth}$ | 0.7965 | 0.8163 | 0.8240 | 0.8336 | 0.8456 | 0.8506 |
| Swin+Smooth | 0.7949 | 0.7958 | 0.7925 | 0.8176 | 0.8351 | 0.8346 |
| Swin+Relabel | 0.7511 | 0.7647 | 0.7643 | 0.7727 | 0.7941 | 0.8087 |
| C-Tran+Relabel | 0.6918 | 0.7172 | 0.7215 | 0.7463 | 0.7686 | 0.7772 |
| Swin+Relabel+TkLR-No-Rank | 0.7679 | 0.7747 | 0.7761 | 0.7872 | 0.8000 | 0.8054 |
| **Swin+Relabel+TkLR** | **0.8143** | **0.8249** | **0.8296** | **0.8399** | **0.8468** | **0.8545** |
| Methods | CUB-200-2011 | | | Indoor67 | | |
| | NDCG@3 | NDCG@5 | NDCG@7 | NDCG@3 | NDCG@5 | NDCG@7 |
| Swin+CE | 0.9025 | 0.9120 | **0.9159** | 0.8987 | 0.9076 | 0.9119 |
| Swin+Top-k SVM | 0.8860 | 0.8987 | 0.8991 | 0.8749 | 0.8841 | 0.8859 |
| Swin+Top-k Entropy | 0.7170 | 0.8294 | 0.8550 | 0.8518 | 0.8352 | 0.8318 |
| Swin+Top-k SVM$_{Semismooth}$ | 0.8937 | 0.9018 | 0.9022 | 0.8868 | 0.8939 | 0.8923 |
| Swin+Smooth | 0.9011 | 0.9056 | 0.9071 | 0.8811 | 0.8853 | 0.8909 |
| Swin+Relabel | 0.8463 | 0.8602 | 0.8754 | 0.8617 | 0.8799 | 0.8817 |
| C-Tran+Relabel | 0.8247 | 0.8307 | 0.8460 | 0.8177 | 0.8315 | 0.8292 |
| Swin+Relabel+TkLR-No-Rank | 0.8704 | 0.8704 | 0.8831 | 0.8734 | 0.8731 | 0.8796 |
| **Swin+Relabel+TkLR** | **0.9110** | **0.9151** | 0.9152 | **0.9016** | **0.9100** | **0.9137** |

## 4.4 EXPERIMENTAL RESULT ON IMAGE DATASET

Table 4 and Table 5 show the top-$k$ accuracy and the $NDCG$ values for different $k$ of all methods on image dataset, respectively. The experimental results on the image dataset are similar to those on the text dataset, our method achieves the best results overall on both top-$k$ accuracy and $NDCG$ metrics, which illustrates the generality of our method.

Compared with the Swin+CE method, our method Swin+Relabel+TkLR has improved the top-$k$ accuracy, especially on the dataset with low top-$k$ accuracy such as Cifar100 and Aircraft. Although Swin+CE performs well on the $NDCG$ metric due to the cross entropy, and gets slightly better results than Swin+Relabel+TkLR at $NDCG@7$ on CUB-200-2011, Swin+Relabel+TkLR still gets the best results on the $NDCG$ metric on all other cases. This shows that the *rank loss* is still very effective on image datasets. The results between Swin+CE, Swin+Top-k SVM, Swin+Top-k SVM$_{Semismooth}$ and Swin+Smooth are almost similar and better than Swin+Top-k Entropy. But the performance of Swin+CE on the $NDCG$ metric is better, because other methods neglect the ranking of ground truth label. The experimental performance of C-Tran+Relabel is poor on two metrics, especially on datasets Cifar100. Compared with the method Swin+Relabel+TkLR-No-Rank, Swin+Relabel+TkLR obviously has a better performance both on top-$k$ accuracy and $NDCG$ metric, which shows the *rank loss* could also improve the top-$k$ accuracy and the $NDCG$ metric on the image dataset.

## 5 CONCLUSION

In this paper, we propose the importance of ground truth label ranking in top-$k$ classification, which none of the previous top-$k$ classification methods take into account. Inspired by the multi-label classification, we transform the top-$k$ classification problem into a problem of predicting the most likely $k$ labels and we propose a novel three-stage approach for top-$k$ classification, which can be easily deployed with different classification models. We then propose an ensemble based relabeling method and relabel the training data with $k$ labels. Besides, we propose a novel top-$k$ loss function that takes the ranking of the ground truth label into account. Finally, we conduct extensive experiments on both text datasets and image datasets, and the experimental results show that our method achieves the best results.

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

## A  APPENDIX

### A.1  THEORETICALLY ANALYSIS OF HYPERPARAMETERS FOR ENSEMBLE LEARNING

In statistics and machine learning, ensemble learning uses multiple learning algorithms to obtain better predictive performance than could be obtained from any of the constituent learning algorithms alone. Take the easy binary classification as an example. Given $y \in \{-1, +1\}$ and the ground truth function $f$. Suppose that each base classifier $h_i$ has an independent generalization error $\epsilon$, i.e.,

$$P\left(h_i(\boldsymbol{x}) \neq f(\boldsymbol{x})\right) = \epsilon.$$

After combining $m$ number of such base classifiers according to

$$H(x) = \text{sign}\left(\sum_{i=1}^{m} h_i(x)\right),$$

the ensemble $H$ makes an error only when at least half of its base classifiers make errors. Therefore, by **Hoeffding inequality**, the generalization error of the ensemble is

$$P(H(x) \neq f(x)) = \sum_{k=0}^{\lfloor m/2 \rfloor} \left(\begin{array}{c} m \\ k \end{array}\right) (1 - \epsilon)^k \epsilon^{m-k} \leq \exp\left(-\frac{1}{2}m(2\epsilon - 1)^2\right)$$

The above formula shows that when the number $m$ of classifiers in the ensemble increases, the error rate of the ensemble decreases exponentially. That is to say, a higher number of classifiers can make the prediction results of the ensemble model more accurate.

In our method, the basic classifier in stage 1 is used to predict the $k$ most likely labels for samples, the prediction results is extremely important for subsequent training. Using only one basic classifier cannot guarantee the stability and accuracy of the prediction results, so we use the idea of ensemble learning to train multiple classifiers in stage 1. However, the number of ensemble models is not as good as possible, too many models will consume a lot of resources and time. At the same time, when the number of models reaches a certain level, increasing the number of models has little effect on the prediction results. At the same time, in order to ensure the diversity of each basic classifier,

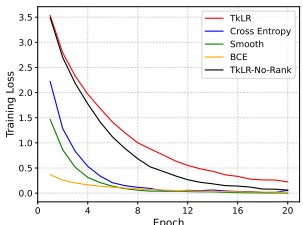 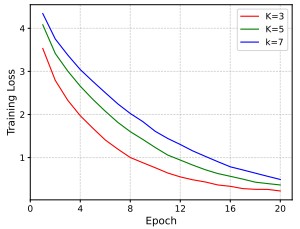 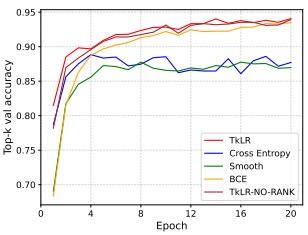

(a) Training loss of different loss functions   (b) Training loss of TkLR for different $k$   (c) Validation accuracy of different loss functions

Figure 3: Training process

the training data of each classifier should be different, so $\alpha$ cannot be 1. However, if $\alpha$ is too small, a better basic classifier cannot be trained. Therefore, in practice, the number $m$ of ensemble models and the proportion $\alpha$ are usually determined through experiments such as Fig.2.

In our experiments, we employ Bagging method to train multiple base classifiers. The Bagging method randomly selects data with a proportion of $\alpha$ from the total training data for training each time. If $\alpha$ is too small, the classifier will underfit, and if $\alpha$ is too large, the classifiers will be too similar, both of which will make the prediction performance of the ensemble classifiers worse. Thus, $\alpha$ is usually obtained through experimental verification. We conduct experiments both on text and image datasets to determine the number of base classifiers $m$ and proportion $\alpha$. We first randomly initialize $\alpha$, and determine the optimal $m$ with the fixed $\alpha$. Then determine $\alpha$ with the optimal $m$. The experimental results are shown in Figure 2. Figure 2a and Figure 2b show that in general, the larger $m$ is, the higher the top-$k$ accuracy is. On the Ohsumed dataset, when $m = 8$, the overall results are good, and the performance is no longer significant when $m$ continues to increase. On the Cifar100 dataset, the result is also good when $m = 5$, and the effect of increasing $m$ is not obvious. Figure 2c and Figure 2d show the performance of different $\alpha$ with the optimal $m$. The results show that when $\alpha = 0.8$, top-$k$ classifier can achieve the best performance on the Ohsumed dataset and the cifar100 dataset. We found similar experimental results on other datasets. Therefore, for hyperparameter $m$, we set the $m$ to 8 for the text dataset and $m$ to 5 for the image dataset. For hyperparameter $\alpha$, we set $\alpha$ to 0.8 for both the text dataset and the image dataset.

### A.2 CHARACTERISTICS INVESTIGATION OF THE LOSS FUNCTION

**Time complexity.** We will discuss the time complexity of our loss function in this part. We first ignore the label ranking part and start with $k = 1$, then our loss function is define as:

$$\mathcal{L}_{k=1} = \log\left(1 + \sum_{n\in\boldsymbol{neg}} e^{s_n - s_p}\right)$$

which is equal to cross-entropy. Obviously, the time complexity of the loss function is $O(1)$. Suppose the total number of categories is $c$, when $k > 1$, the complexity of Eq.4 depends on

$$\sum_{n\in\boldsymbol{neg},p\in\boldsymbol{pos}} e^{s_n - s_p}$$

where $|\boldsymbol{pos}| = k$ and $|\boldsymbol{neg}| = c - k$. The total number of calculations is $k(c-k)$. Due to the number of categories $c$ is much larger than $k$ in practice, the complexity of Eq.4 is $O(c * k)$. At the same time, since the complexity of the label ranking part is $O(1)$, the time complexity of the loss function is $O(c * k)$.

**Training process.** To illustrate the performance of our loss function, we show the training process of our loss function. Fig.3 shows the training process. First of all, it can be seen from Fig.3a that our loss function converges slower than other loss functions, and the initial loss is larger. Comparing TkLR and TkLR-NO-RANK at the same time, it is obvious that TkLR-NO-RANK converges faster, which shows that label ranking has an impact on the function. At the same time, TkLR-NO-RANK can converge to a loss value close to 0 like other functions, which shows that our proposed loss

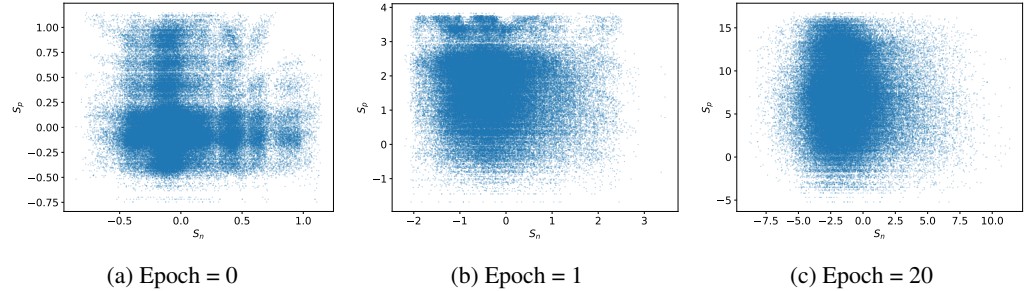

(a) Epoch = 0          (b) Epoch = 1          (c) Epoch = 20

Figure 4: Visualization of top-$k$ classification on test set. Top-$k$ classification maximizes $s_p - s_n$. Fig (a) and (b) are the unconverged state, and Fig (c) is the converged state.

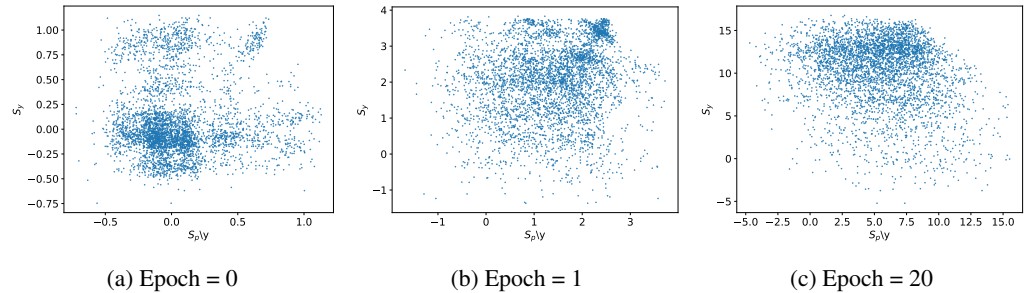

(a) Epoch = 0          (b) Epoch = 1          (c) Epoch = 20

Figure 5: Visualization of label ranking on test set. Label ranking maximizes $s_y - s_p \backslash y$, where $\backslash y$ means $p \neq y$. Fig (a) and (b) are the unconverged state, and Fig (c) is the converged state.

function is effective. Fig3b shows the training process for different $k$. It can be seen from the figure that the larger $k$ is, the slower the convergence is. Due to the complexity of the loss function is $O(c * k)$, the larger $k$ is, the greater the loss is. Fig.3c shows the top-$k$ accuracy on the validation set during training with different loss functions. It can be seen from the figure that our loss function not only achieves better results, but also leads other loss functions at the beginning of training. The results of the multi-label methods are significantly better than the single-label methods, which shows the effectiveness of our idea of using multi-label to solve top-k problem.

**Analysis of the convergence.** We illustrate the state of convergence by analyzing the two objectives of the loss function. The first objective is top-$k$ classification which aims to maximize $(s_p - s_n)$. Fig.4 shows the state of $s_p$ and $s_n$ from never converged to converged. Fig.4a shows the distribution of $s_p$ and $s_n$ when epoch=0 (training is not started). From the figure, it can be seen that $s_p$ and $s_n$ are distributed near (0,0). When epoch=1, Fig.4b shows that the distribution slowly moves to the left and up, which indicates that the objective is converging. When epoch=20, Fig.4c shows that the distribution of $s_p$ is concentrated in [0, 15], while the distribution of $s_n$ is concentrated in [-5, 2.5], which is in line with the goal of maximizing $(s_p - s_n)$. At the same time, compared with Fig.4a and Fig.4b, the distribution of Fig.4c is more concentrated, which shows that the convergence of the objective function is completed.

Another objective is label ranking which aims to maximize $(s_y - s_p), p \neq y$. Fig.5 also shows the state of $s_y$ and $s_p$ from never converged to converged. Fig.5a shows that the distribution of $s_y$ and $s_p \backslash y$ is concentrated around (0,0) when the initial state epoch=0. Fig.5b shows that the distribution of $s_y$ and $s_p \backslash y$ shifts upward when epoch=1, which shows that the optimization objective $(s_y - s_p)$ is converging. Fig.5c shows that $s_y$ is distributed between [8, 15], and $s_p \backslash y$ is distributed between [0, 10]. Compared with Fig.5a and Fig.5b, the distribution in Fig.5c is more concentrated, and the objective is greatly optimized, so the convergence is complete.

