# OpenReview forum: "Learning Top-k Classification with Label Ranking"
_ICLR.cc/2023/Conference — Submitted to ICLR 2023_

### Official Review · Reviewer_ALCH · 2022-10-21

**Confidence:** 3
**Correctness:** 3
**Technical Novelty And Significance:** 2
**Empirical Novelty And Significance:** 2
**Recommendation:** 5

**Clarity, Quality, Novelty And Reproducibility:**

Basically, I like the paper's motivation; it is quite a fundamental research problem to learn top-k classification with label ranking. Besides, this paper is well-structured, and easy to follow. I appreciate that the paper conducted experiments based on many real datasets. The experimental result shows that the proposed three-stage approach can more effectively perform top-k classification than the previous approaches.

Despite its practical usefulness, the proposed approach seems to be straightforward. I think that more technical depth is required. As for two hyperparameters of m and alpha (the number of base classifiers and proportion), the paper concludes that m should be set to 8 for the text dataset and 5 for the image dataset, and alpha should be set to 0.8 from the experimental results. I think it is good to theoretically analyze the impact of hyperparameters for the proposed approach to improve the technical depth of the paper.

I appreciate the generality of the proposed approach; it can use various machine learning approaches in performing top-k classification. However, the paper tested only BERT, LSAN, Swin, and C-Tran in the experiment. I think, to show the generality of the proposed approach, more machine learning approaches should be used in the experiments, such as SVM, which is a popular approach for top-k classification, as described in Section 2.

**Strength And Weaknesses:**

Strength:
- Simple model to be easily implemented.
- General approach to accommodate various machine learning approaches.
- Extensive experiments are performed in the paper.

Weakness:
- Technical depth of the proposed approach is not so deep.
- More machine learning approaches should be tested to evaluate the effectiveness of the proposed approach used in the proposed approach.
- The impact of hyperparameters should be theoretically analyzed.

**Summary Of The Paper:**

This paper proposes an approach that performs top-k classification with label ranking. The proposed approach has three stages. The first stage trains the base classifier. The second stage relabels training data using one ground truth label and k-1 most likely labels. The third stage trains the top-k classification model by introducing a new top-k loss function. Using text and image datasets, the paper evaluated the effectiveness of the proposed approach.

**Summary Of The Review:**

The proposed approach is technically somewhat shallow.
Theoretical analysis for hyperparameters should be added to the paper.
To show the generality of the proposed approach, more machine-learning approaches should be tested in the experiment.

---

> ### Author Response · Authors · 2022-11-18
> **Response**
>
> **Comment 4.1**. Despite its practical usefulness, the proposed approach seems to be straightforward. I think that more technical depth is required. As for two hyperparameters of m and alpha (the number of base classifiers and proportion), the paper concludes that m should be set to 8 for the text dataset and 5 for the image dataset, and alpha should be set to 0.8 from the experimental results. I think it is good to theoretically analyze the impact of hyperparameters for the proposed approach to improve the technical depth of the paper.
> **Response 4.1**: Thanks for your suggestion. We have added the theoretically analysis of hyperparameters in Appendix A.1.
>
> **Comment 4.2**. I appreciate the generality of the proposed approach; it can use various machine learning approaches in performing top-k classification. However, the paper tested only BERT, LSAN, Swin, and C-Tran in the experiment. I think, to show the generality of the proposed approach, more machine learning approaches should be used in the experiments, such as SVM, which is a popular approach for top-k classification, as described in Section 2.
> **Response 4.2**: Thanks for your suggestion. We have added 3 comparison method, including two top-k SVM based methods in the experiment. Experimental results show that our method achieves the best results.

---

### Official Review · Reviewer_iYfY · 2022-10-23

**Confidence:** 4
**Correctness:** 3
**Technical Novelty And Significance:** 3
**Empirical Novelty And Significance:** 2
**Recommendation:** 6

**Clarity, Quality, Novelty And Reproducibility:**

Overall the paper is clear and well-written. It is novel at large. The descriptions of the experiments are generally detailed, so the reproducibility should not be a concern. It would be better to include the code.

**Details Of Ethics Concerns:**

Not applicable.

**Strength And Weaknesses:**

**Strong points.**

The consideration of the ranking of the ground truth label is well-motivated and is of significance for top-k classification problems. The proposed 3-stage method is relatively intuitive, and was clearly described. The experimental results show the method is very effective.

**Weak points.**

The proposed 3-stage method is more complex and requires more computations. It is not clear from the experiments whether the better performances were due to the more complex model and more computations. Comparing with other ensemble methods might help make this point clearer. Nevertheless, it seems that the new loss function in the third stage did help improve performances. More theoretical understanding of the proposed loss function is needed.

**Summary Of The Paper:**

The paper studies top-k classification problem. Observing existing top-k classification methods neglect the ranking of the ground truth label among the predicted k labels, the authors proposed to take label ranking into consideration. Specifically, a new 3-stage approach was proposed to tackle this problem. In the first stage, an ensemble method is to train several base classifiers. In the second stage, these base classifiers are used to predict class probabilities, which are then averaged to create top-k most likely labels (including the ground truth label) for each example. In the third stage, a novel loss function that considers the ranking of the ground truth label is used to train a top-k classifier. The experimental results show that the performances of the proposed method are significantly better than those of existing methods on several text and image datasets.

**Summary Of The Review:**

I vote for a weak accept since I think the merits outweigh the weak points.

**Questions.**

1. How many trials were conducted in the experiments?

**Comments.**

1. In Eq. 7, "pso" -> "pos".

---

> ### Author Response · Authors · 2022-11-18
> **Response**
>
> **Comment 3.1**: The proposed 3-stage method is more complex and requires more computations. It is not clear from the experiments whether the better performances were due to the more complex model and more computations. Comparing with other ensemble methods might help make this point clearer. Nevertheless, it seems that the new loss function in the third stage did help improve performances. More theoretical understanding of the proposed loss function is needed.
> **Response 3.1**:  Thanks for the comments. Actually, we have tested different number of base classifiers for ensemble stage, and the results show that more base classifiers (i.e., more computations) can provide more accurate results and when the base classifier number exceeds a certain threshold, the top-k accuracy becomes relatively stable. In the meantime, Ensemble learning is indeed helpful to our method, but compared with other methods (BERT+SMooth and BERT+BalancedLoss) that also use Ensemble learning, our method can still achieve better results, which reflects the superior performance of our loss function. Finally, we have added some theoretical analysis for the loss function in Appendix A.2, including the training process and convergence analysis.
>
> **Comment 3.2**.  How many trials were conducted in the experiments?
> **Response 3.2**: Thanks for your question. In order to facilitate the reproduction of the paper, we fixed the random seed in the experiment. At the same time, we changed different random seeds and conducted three trials, and the results of each trial showed that our method significantly outperforms other, so we only report the results of one random trial with same seed.

---

### Official Review · Reviewer_Pewp · 2022-10-25

**Confidence:** 4
**Correctness:** 3
**Technical Novelty And Significance:** 2
**Empirical Novelty And Significance:** 3
**Recommendation:** 5

**Clarity, Quality, Novelty And Reproducibility:**

As mentioned before, the clarity and the writing quality are good. I am not sure about the paper's novelty though. The key technical contribution, IMHO, is the introduction of the new loss function for training the top-k classifier. It is not a sophisticated change, and more discussion on training with this new loss could make the paper better.

**Strength And Weaknesses:**

Strength:
- The writing of this paper is clear. The proposed idea is presented in a way that even for readers not in the field could follow easily.
- The empirical study has considered both text and image datasets, with different characteristics. And the proposed approach has produced reasonable results.

Weaknesses:
- I think the proposed setting, order-aware top-k classification, should be better motivated.
- One of the key technical contributions is the loss function used to train the top-k classifier. More discussion about this loss function is desired.
- I think the related work can also be strengthened.

**Summary Of The Paper:**

This paper studies the top-k classification problem. Different from the existing top-k classification setting, in this paper the labels (including the ground truth label) are ranked. It proposes a three-stage approach to this problem:
1. base classifier construction,
2. training data construction with relabeling, and
3. training top-k classifier with a revised order-sensitive loss function. The proposed approach is tested on text and image datasets.

**Summary Of The Review:**

- The motivating example in the introduction doesn't properly highlight the setting of the paper. Indeed that in the human-in-the-loop scenario,  we would like to present the annotators the most relevant data for annotation; But in this scenario, isn't the setting the same as retrieval or, even more so, label ranking, where only top-k ranked items are shown? I think coming up with an appealing motivating example is important, especially as the experiments are based on syntactic datasets. (By which I mean the datasets are augmented to study the setting.)

- The setting considered in this paper is in fact exactly the same as the one in *label ranking*, when the evaluation focuses on top-ranked label(s). The authors should consider them in the related work.

- Between Eq. (4) and (5): "should **be** as large as possible".

- I was expecting more discussion about the modified loss function in Eq. (5). For example, when you put it on top of BERT, what does the training process look like? Does it converge fast?

- Experiments. The results of LSAN and C-Tran are so underwhelming, that one would ask why they were included and why the authors called them state of the art for the respective tasks. No better baseline could be found?

---

> ### Author Response · Authors · 2022-11-18
> **Response**
>
> **Comment 2.1.**  The motivating example in the introduction doesn't properly highlight the setting of the paper. Indeed that in the human-in-the-loop scenario, we would like to present the annotators the most relevant data for annotation; But in this scenario, isn't the setting the same as retrieval or, even more so, label ranking, where only top-k ranked items are shown? I think coming up with an appealing motivating example is important, especially as the experiments are based on syntactic datasets. (By which I mean the datasets are augmented to study the setting.)
> **Response 2.1**: Thanks for your suggestion. In the human-in-the-loop motivation scenario, the result of top-k classification has only one ground truth label, and the other labels are all wrong. Although our work ranks k results, we only care about the ranking of ground truth label, that is, to improve the ranking of the ground truth label, and we don't care about the ranking of other wrong results. Improving the ranking of the ground truth label in the k labels will allow humans to get the ground truth label at the first time, which could effectively improve the efficiency of humans checking. Therefore, this scenario is different from scenarios such as information retrieval.
>
> **Comment 2.2**.  The setting considered in this paper is in fact exactly the same as the one in label ranking, when the evaluation focuses on top-ranked label(s). The authors should consider them in the related work.
> **Response 2.2**:  Thanks for your suggestion. As mentioned before, the goal of our method is different from label ranking, we only consider the ranking of the ground truth label. Label ranking is only used to improve the ranking of the ground truth label in the predicted k labels. In fact, in Lapin's [1] work, top-k SVM has been compared with label ranking. The experimental results show that label ranking is not effective in solving top-k classification.
> [1] Lapin, Maksim . Top-k multiclass SVM. In NIPS 2015.
>
> **Comment 2.3**.  Between Eq. (4) and (5): "should be as large as possible".
> **Response 2.3**:  Thanks for your suggestion.  We have revised this.
>
> **Comment 2.4**.  I was expecting more discussion about the modified loss function in Eq. (5). For example, when you put it on top of BERT, what does the training process look like? Does it converge fast?
> **Response 2.4**:  Thanks for your question. We have added a discussion section of the loss function in the Appendix A.2, including the training process, convergence analysis.
>
> **Comment 2.5**. Experiments. The results of LSAN and C-Tran are so underwhelming, that one would ask why they were included and why the authors called them state of the art for the respective tasks. No better baseline could be found?
> **Response 2.5**: Thanks for your comment. We have replaced LSAN with a relatively recent method, named BalancedLoss [1]. In fact, we have also found another SOTA method MrMP [2]  that can be modified to solve our problem, but it performs even worse than BalancedLoss in our problem setting, so we didn’t report its results. As for C-Tran,considering it performs relatively well in some datasets, we retained its results in the revision.
> [1]  Yi Huang, Buse Giledereli, Abdullatif Koksal, Arzucan Ozgur, and Elif Ozkirimli. Balancing methods for multi-label text classification with long-tailed class distribution. In EMNLP 2021.
> [2] M. Ozmen, H. Zhang, P. Wang, and M. Coates, “Multi-relation message passing for multi-label text classification,” In ICASSP 2022.

---

### Official Review · Reviewer_6iPJ · 2022-10-25

**Confidence:** 4
**Correctness:** 3
**Technical Novelty And Significance:** 1
**Empirical Novelty And Significance:** 2
**Recommendation:** 5

**Clarity, Quality, Novelty And Reproducibility:**

The paper is overall well-presented, but the novelty is not enough. Making the top-k classification a form of multi-label learning with the groundtruth class being encouraged is interesting but incremental.

**Strength And Weaknesses:**

Pros:
1. The proposed method is simple and easy to follow.
2. The experiments show that the proposed method outperforms other baselines.

Cons:
1. My first concern is that training base classifiers to obtain the prediction uncertainty is time-consuming which is however not discussed in the paper.
2. The three-stage approach which relabels the training data with multiple labels and enhances the ground-truth one is reasonable but not novel. The formulated objective, i.e., Eq. (5), is correct but is not very inspiring to me.
3. Following 1&2, one can alternatively do dropout at stage 1 to have the uncertainty measure.
4. The noisy label was mentioned in the introduction as a motivation but was not discussed later.


**Summary Of The Paper:**

This paper proposes a three-stage approach to learning top-k classification with label ranking concerned during modeling.

**Summary Of The Review:**

This work has not touched the core problem of top-k classification (e.g., non-differentiable) and some related (e.g., Sparsemax+top-K truncation) works are absent.

Solving top-k classification from the view of multi-label perspective is interesting to some audiences but perhaps needs more technical supports.

---

> ### Author Response · Authors · 2022-11-18
> **Response**
>
> **Comment 1.1**. My first concern is that training base classifiers to obtain the prediction uncertainty is time-consuming which is however not discussed in the paper.
> **Response 1.1**: Thanks for your comment. In stage 1, it is true that obtaining the prediction uncertainty is time-consuming. However, the base classifiers in this stage can be trained in parallel manner, so it doesn't take too much time in real scenario as long as we can train different base classifiers in different machines.
>
> **Comment 1.2**. The three-stage approach which relabels the training data with multiple labels and enhances the ground-truth one is reasonable but not novel.
> **Response 1.2**: Thanks for your comment. Although the relabeling process is relatively simple, it is indeed very effective in our method, and no one has tried it in top-k classification.
>
> **Comment 1.3**. Following 1&2, one can alternatively do dropout at stage 1 to have the uncertainty measure.
> **Response 1.3**: Thanks for your comment. Although do dropout can obtain the uncertainty measure, its prediction accuracy cannot be guaranteed. Ensemble learning is more capable of obtaining more accurate prediction results, which is helpful for subsequent training. Therefore we adopt ensemble learning in the paper.
>
> **Comment 1.4**. The noisy label was mentioned in the introduction as a motivation but was not discussed later.
> **Response 1.4**: Thanks for your comment. Noisy label is not the key point of this paper, but our method is also applicable to the situation with noisy labels. Our method could recommend the k most likely labels, even if the label of the sample is incorrect, we still have a high probability of getting the correct label from the k labels.
>
> **Comment 1.5**. This work has not touched the core problem of top-k classification (e.g., non-differentiable) and some related (e.g., Sparsemax+top-K truncation) works are absent.
> **Response 1.5**: Thanks for your comment. Previous work tackles the problem from a non-differentiable perspective, which is just one way to tackle top-k classification. Our method solves the top-k classification from a different perspective, i.e., to utilize the multi-label nature of samples for top-k classification. To the best of our knowledge, no one has solved the top-k classification problem from a multi-label perspective. Hence, we think our work is novel. More importantly, we have added more comparison methods (non-differentiable based top-k classification methods, e.g., top-k SVM, top-k Entropy and so on) for experimental evaluation in this revision, our work is very effective for top-k classification and the experiments show that our method significantly outperform all other existing methods that we can find.

---

> > ### Comment · Reviewer_6iPJ · 2022-12-13
> > **Comments after reading the response**
> >
> > I appreciate the authors' response. After reading all the comments and authors' responses, I believe the authors have put many efforts on this work. Unfortunately, I cannot accept this paper right now, because this work has not reach the average level of all my reviewed ICLR papers from the view of novelty and technical contribution.

---

### Decision · Program_Chairs · 2023-01-20

**Decision:**

Reject

**Justification For Why Not Higher Score:**

The paper received only one score above a bar (6) and the rest of scores were (5). After reading the paper by myself, I found that the paper has several flaws. The most important ones I have mentioned in my meta-review. I did not see a need of having an AC-reviewer meeting as the flaws are too serious to consider this paper for acceptance.

**Justification For Why Not Lower Score:**

N/A

**Metareview: Summary, Strengths And Weaknesses:**

The paper considers a specific instance of multi-class classification based on sorted top-k predictions. The authors introduce a method that reduces the problem to multi-label classification. In the first step multi-label examples are obtained by using predictions of an ensemble of multi-class classifiers. In the next step a top-k multi-label classifier is used. Experimental studies are used for verification of the proposed approach.

The considered problem is not really well-defined. It is not clear what task loss should be minimized and what optimal prediction in this setting is. Top-k and NDCG performance metrics used in experimental studies are relatively different from each other (please note that the optimal solution for NDCG is not a ranking according to conditional probabilities of labels). In general, the paper lacks theoretical insights to the considered problem.

The proposed method is quite complex and very adhoc without any theoretical analysis. We do not know how close we can get with the proposed approach to the optimal solution (assuming that it was properly defined). Also the empirical studies seem to be a bit unfair. For example, the cross-entropy approaches do not use any kind of ensembling which might result with lower performance in comparison to the introduced method.

Such a submission needs more solid theoretical grounding to justify the presented ideas. Therefore, the paper at the current stage is not ready for publication at the top conference in machine learning.

**Summary Of Ac-Reviewer Meeting:**

N/A